# Association Between Microcalcification Patterns in Mammography and Breast Tumors in Comparison to Histopathological Examinations

**DOI:** 10.3390/diagnostics15131687

**Published:** 2025-07-02

**Authors:** Iqbal Hussain Rizuana, Ming Hui Leong, Geok Chin Tan, Zaleha Md. Isa

**Affiliations:** 1Department of Radiology, Faculty of Medicine, Universiti Kebangsaan Malaysia Medical Center, Jalan Yaakob Latif, Kuala Lumpur 56000, Malaysia; rizi7886@gmail.com (I.H.R.); minghui1115@gmail.com (M.H.L.); 2Hospital Canselor Tuanku Muhriz, Jalan Yaakob Latif, Bandar Tun Razak, Kuala Lumpur 56000, Malaysia; 3Department of Pathology, Faculty of Medicine, Universiti Kebangsaan Malaysia Medical Center, Jalan Yaakob Latif, Kuala Lumpur 56000, Malaysia; 4Department of Community Health, Faculty of Medicine, Universiti Kebangsaan Malaysia Medical Center, Jalan Yaakob Latif, Kuala Lumpur 56000, Malaysia

**Keywords:** mammogram, microcalcifications, mass, histopathology, breast malignancy, ductal carcinoma in situ, invasive carcinoma

## Abstract

**Background/Objectives:** Accurately correlating mammographic findings with corresponding histopathologic features is considered one of the essential aspects of mammographic evaluation, guiding the next steps in cancer management and preventing overdiagnosis. The objective of this study was to evaluate patterns of mammographic microcalcifications and their association with histopathological findings related to various breast tumors. **Methods:** 110 out of 3603 women had microcalcification of BIRADS 3 or higher and were subjected to stereotactic/ultrasound (USG) guided biopsies, and hook-wire localization excision procedures. Ultrasound and mammography images were reviewed by experienced radiologists using the standard American College of Radiology Breast-Imaging Reporting and Data System (ACR BI-RADS)**. Results**: Our study showed that features with a high positive predictive value (PPV) of breast malignancy were heterogeneous (75%), fine linear/branching pleomorphic microcalcifications (66.7%), linear (100%), and segmental distributions (57.1%). Features that showed a higher risk of association with ductal carcinoma in situ (DCIS) were fine linear/branching pleomorphic (odds ratio (OR): 3.952), heterogeneous microcalcifications (OR: 3.818), segmental (OR: 5.533), linear (OR: 3.696), and regional (OR: 2.929) distributions. Furthermore, the features with higher risks associated with invasive carcinoma had heterogeneous (OR: 2.022), fine linear/branching pleomorphic (OR: 1.187) microcalcifications, linear (OR: 6.2), and regional (OR: 2.543) distributions. The features of associated masses in mammograms that showed a high PPV of malignancy had high density (75%), microlobulation (100%), and spiculated margins (75%). **Conclusions:** We concluded that specific patterns and distributions of microcalcifications were indeed associated with a higher risk of malignancy. Those with fine linear or branching pleomorphic and segmental distribution were at a higher risk of DCIS, whereas those with heterogeneous morphology with a linear distribution were at a higher risk of invasive carcinoma.

## 1. Introduction

According to the Malaysia National Cancer Registry Report 2012–2016, breast cancer is the most common form of cancer affecting women in Malaysia, with an age-standardized incidence rate (ASR) as high as 34.1 per 100,000—this makes it three times more common than colorectal cancer (with an ASR of 11.1), the second most common form of cancer. Most breast cancer cases in Malaysia are diagnosed at stage II, accounting for 37% of all new cases of breast cancer detected from 2007 to 2011 [1]. Many breast cancer patients experience a decrease in quality of life. This disease can have significant psychological impacts and can induce stress in patients; it is life-threatening, and mastectomies, chemotherapy effects, and disrupted social and daily living activities can cause body image issues. Furthermore, breast cancer can also cause psychological distress or even depression in the patient’s caregiver [2,3,4].

Since the first description of microcalcifications in mastectomy specimens by Solomon in 1913, there were many studies on the relationship and association between microcalcifications and breast diseases and the importance of the radiological–pathological correlation in diagnosing these diseases; this is especially true when assessing radiologically detected microcalcifications in surgically removed breast specimens, which became apparent in the 1970s. The introduction of mammography screening substantially increased the number of diagnosed microcalcification cases and led to physicians realizing their importance in detecting breast carcinomas early [5].

Mammograms remain the primary screening tool for breast cancer, as recommended by the American College of Radiology and Clinical Practice Guideline (CPG) Malaysia [6,7,8]. According to CPG Malaysia, a screening mammogram is recommended every other year for women aged 50–74 with no risk factor. For women with a high breast cancer risk factor but in whom no genetic variants were identified, an annual screening mammogram is recommended from ages 40 to 59 and every other year after 60 [6]. At Hospital Canselor Tuanku Muhriz (HCTM), complementary Ultrasounds (USGs) are performed with mammogram studies. Studies showed that combining digital breast tomosynthesis mammograms with USGs can significantly improve diagnostic sensitivity [9].

Over the years, more women in Malaysia enrolled in the mammography screening program; hence, the identification and investigation of microcalcifications found in mammography became more common [10]. In some cases, women may not benefit from screening programs because mammogram calcifications might be over-diagnosed. Several studies in the literature report an overdiagnosis rate for mammographic calcifications ranging from 0 to 54% [11,12].

There are several manifestations of breast carcinoma in mammography, such as ill-defined, spiculated masses, and clustered, parallel linear, or irregular intraductal calcification microcalcifications [13,14]. Microcalcifications in mammograms can exhibit many subtle variations in number, size, shape, extent, density, and distribution patterns. Careful examination is needed to detect mammographic lesions and calcifications; categorize them as benign, suspicious, or malignant; and recommend proper surgical interventions and prognostication [15]. Therefore, accurately correlating mammographic findings with corresponding histopathologic features is considered one of the essential aspects of mammographic evaluation [13]. The fifth edition of the BI-RADS lexicon, published in 2013, describes breast microcalcifications and provides diagnostic categories with standardized biopsy recommendations. Despite this, linking descriptive findings with assessment categories is still a challenging diagnostic task in evaluating microcalcifications.

## 2. Materials and Methods

### 2.1. Study Design

This single-centered retrospective study was carried out at Hospital Canselor Tuanku Muhriz UKM (HCTM). It was presented to the HCTM ethical review board and was approved.

### 2.2. Study Population

A total of 3603 women underwent digital breast tomography mammogram examinations from 1 January 2020 to 31 December 2021. Out of 3603, only 110 women had mammograms showing microcalcifications. A proportion of these women were symptomatic (29/110, 26.4%). They subsequently underwent stereotactic/USG-guided core biopsies or hook-wire localization excision removal of microcalcification procedures, and they were included in this study.

### 2.3. Study Methodology

Stereotactic/USG-guided biopsies or hook-wire localization excision removal of microcalcification procedures performed at HCTM were retrieved from the procedure registration book. The data of patients who fit the inclusion criteria were collected via the Integrated Laboratory Management System (ILMS) software, the Integrated Radiology Information System (IRIS) software, and the HCTM Caring Hospital Enterprise System (CHETS) from 2020 to 2021. Ultrasound and mammography images were traced from the picture archiving and communicating system (PACS) and reviewed; the imaging findings were characterized by radiologists with more than 10 years of experience with special interest in breast imaging and reading mammograms according to the American College of Radiology Breast-Imaging Reporting and Data System (ACR BI-RADS) Atlas, 5th Edition. Patient records were made anonymous and deidentified before analysis.

### 2.4. Consent

Patient consent was waived as the images were acquired retrospectively and did not impact the management of the patients.

### 2.5. Statistical Methods

Statistical analysis was performed using commercially available software (SPSS version 26.0). Pearson’s chi-squared test was used to assess the statistical significance of differences between categorical variables.

## 3. Results

### Definition

Microcalcification morphologies and distributions were defined following the ACR BI-RADS lexicon, as summarized in Table 1 [16,17].

**Table 1 diagnostics-15-01687-t001:** Descriptions of microcalcification morphologies and distributions.

Morphology	Description
Coarse heterogeneous	Irregular and defined calcifications tend to coalesce. Measuring more than 0.5 mm, i.e., more than pleomorphic calcifications but less than dystrophic calcifications.
Amorphous	Alternatively called “powder”, “cloud”, or “cottony”, these correspond to such small calcifications (less than 0.1 mm) that it is not possible to count them or determine their shapes. It is also called tiny or hazy microcalcifications, about 200 to 300 µm, they are less conspicuous than the other microcalcifications and require technically optimized mammograms [18].
Fine Pleomorphic	Alternatively called “crushed stone”, these correspond to calcifications of different shapes and sizes; they are angled and heterogeneous, with a size between 0.5 and 1 mm, smaller than coarse heterogeneous calcifications. It is usually more conspicuous than the amorphous forms of calcifications. Neither typically benign nor typically malignant irregular calcifications with varying sizes and shapes. It is typically more prominent as compared to the amorphous forms. Irregular calcifications of different sizes and shapes that are neither distinctly benign or malignant [19].
Fine linear or branched calcifications	Small calcifications measuring less than 0.5 mm; thin, linear, usually discontinuous, and with irregular edges. They may branch in different directions, forming “letters” (L, V, Y, and X).
**Distribution**	**Description**
Diffuse	Calcifications randomly distributed within the breast.
Group	A few calcifications in a small area, with a lower limit of five calcifications within 1 cm of each other or when there is a definable pattern. The upper limit refers to when there are more microcalcifications present within 2 cm of each other.
Linear	Calcifications arranged in a linear path that can branch.
Segmental	This distribution pattern follows the anatomical shape of a breast lobe, i.e., a triangular shape with the tip directed toward the nipple.
Regional	Calcifications in an extensive area, greater than 2 cm in their largest dimension. Can cover more than one quadrant.

A total of 3603 women underwent mammograms from 1st January 2020 to 31st December 2021, and 110 women had mammograms showing microcalcifications. They subsequently underwent stereotactic/USG-guided core biopsies or hook-wire localization excision removal procedures. Our study comprised women aged 33 to 79 years old, with a mean age of 57.5 ± 10.2. Most of the patients were in the 61–65 age group, followed by the 46–50 age group. Figure 1 shows the distributions of the sample in this study, and Table 2 summarizes the patients’ demographics, presenting symptoms, and types of procedures. Most patients were asymptomatic (*n*-81, 74%), and only 29 (26%) were symptomatic during their mammogram examinations.

The microcalcification types found in the mammograms assessed in this study included fine pleomorphic, coarse heterogeneous, amorphous, and linear branching. The four types of distribution patterns assessed were grouped, segmental, regional, and linear. All microcalcification findings were categorized based on BI-RADS classifications. In total, 2 cases were categorized as BI-RADS 3, 99 cases were categorized as BI-RADS 4, and 9 were cases classified as BI-RADS 5. The prevalence of BIRADS 4 and 5 microcalcifications found in mammograms at HCTM was 2.9%. Histopathological examination (HPE) results based on BI-RADS categories are summarized in Table 3. Of all these lesions, 71 were benign and 39 were malignant. The two BI-RADS 3 lesions were HPE-proven benign lesions. Of the 99 BI-RADS 4 lesions, 31 malignant lesions contributed to a malignancy rate of 31.3%. There was only one HPE-proven benign lesion among the nine BI-RADS 5 lesions; hence, the malignancy rate was 88.9%. Among 3603 patients, only 110 women fulfilled the inclusion criteria of having microcalcifications. Most of these women were classified as BI-RADS 4, while 2 patients fell into the BI-RADS 3 category, and 10 patients fell into the BI-RADS 5 category. The malignancy rate among BI-RADs 5 lesions is lower as compared to the reference malignancy rate (>95%) according to BI-RADS. This discrepancy may have been caused by the small sample size, which limits the representativeness of the findings.

HPE results based on the morphology and distribution of the microcalcifications are summarized in Table 4. No subject presented with microcalcifications with a diffuse distribution.

Based on the morphology and distribution of the microcalcifications, the positive predictive value (PPV) of each parameter in predicting malignancy was calculated, as summarized in Table 5. Table 6 summarizes the association between the morphology and distribution of microcalcifications and breast malignancy.

Every mass associated with microcalcifications found in mammograms was carefully evaluated, followed by a complementary ultrasound. Subsequently, mammographic and sonographic features were described based on the BI-RADS lexicon. For mammographic features associated with breast cancer, we found that only 23.5% of ductal carcinoma in situ (DCIS) had an associated mass in a mammogram, as compared with invasive carcinoma, which was at 47.1%. Spiculated and microlobulation margins were seen in invasive carcinoma but not in DCIS. All the masses found in invasive carcinoma were high-density masses. Based on each characteristic, the PPV of malignancy (given the mammographic mass features) was further divided into mass density and mass margins and calculated, as summarized in Table 7.

Figure 2, Figure 3, Figure 4, Figure 5 and Figure 6 show examples of the different microcalcification morphologies and distributions, while Figure 6 shows an example of a mammographic mass and an image of a complementary ultrasound assessment of the mass.

## 4. Discussion

Mammography is an effective screening test for the timely identification of microcalcifications and can detect 25–43% of non-palpable cancers [20]. In our study, two BI-RADS 3 lesions were HPE-proven benign lesions, consistent with a malignancy rate of <2%, as stated in ACR BI-RADS Atlas 2013. For BI-RADS 4 lesions, the malignancy rate is 33.3%, consistent with a malignancy rate of 2–95%. However, this study’s malignancy rate for BIRADS 5 microcalcifications/lesions was 88.9%, which was slightly lower than the estimated malignancy rate of >95% [16,21]. This lower observed malignancy rate of 88.9% may be attributed to the small sample size, which is a limitation in this study. With a larger sample size, the malignancy rate may closely align with the reference malignancy rates according to the ACR BI-RADS Atlas 2013. One subject with benign histopathology had a BI-RADS 5 lesion. When a benign pathologic test result from a percutaneous biopsy of a BI-RADS 5 lesion is obtained, this is often considered discordant; a repeated biopsy or surgical excision is recommended. However, some benign entities were possibly classified as BI-RADS 5 lesions, including an atypical infection, a complex sclerosing lesion, a radial scar, fat necrosis, fibromatosis, mastitis, and myofibroblastoma [22]. In this study, HPEs showed fibroadenomatous changes.

Fine pleomorphic calcifications were the most common morphology in our study, at 65.4% among all microcalcification types, followed by amorphous calcifications at 10.9%. However, these two microcalcifications had low PPVs in predicting malignancy: 38.9% and 16.7%. The most common HPE result for these two calcifications was benign breast tissue. These microcalcifications were also commonly seen in benign conditions, such as fibrocystic changes. On the other hand, the morphologies associated with a high PPV were heterogeneous and fine linear or branching pleomorphic microcalcifications, at 75.0% and 66.7% (Table 8). However, these findings cannot be generalized due to the small sample size. Further studies involving larger cohorts should be conducted to assess the validity of these findings.

Compared with ACR BI-RADS and other studies, our study found that heterogeneous microcalcification was the only morphology with a high PPV. The other microcalcification morphologies were similar, although Kim et al. described the fine pleomorphic morphology as having a higher PPV (63.2%). For the microcalcification distributions, our study found that the regional distribution had a higher PPV (57.1%) than ACR BI-RADS (26%), which was similar to Liberman et al. (46%), while the PPVs of the other distributions were consistent with other studies [16,23,24].

In our study, five patients had increasing calcifications at 1-to-2-year follow-up intervals; however, none of these were malignant lesions. According to a study by Lev-Toaff et al., the fact that microcalcifications appear or increase does not necessarily indicate malignancy, but if microcalcifications give the impression of suspicious characteristics, regardless of their stability over time, histological studies are necessary. However, Lev-Toaff et al. also concluded that the temporal stability of microcalcifications significantly decreases the risk of invasive carcinoma, but not necessarily of carcinoma in situ [25].

Our study showed that the segmental distribution of microcalcifications had the highest association (OR: 5.5, 95%; CI: 1.146–26.718) with DCIS, followed by the fine linear and branching pleomorphic morphology, with an OR of 3.9 and a 95% CI of 0.744–21.004.

Our study also showed that linear distribution had the highest risk (OR: 6.2, 95% CI: 0.368–104.532) associated with invasive carcinoma, followed by a regional distribution (OR: 2.5, CI 0.449–14.4). Regarding the association between microcalcifications and invasive breast cancer, Lilleborge et al. reported that the fine linear or branching pleomorphic morphology has an OR of 20.0, as compared with fine linear or fine pleomorphic microcalcifications, and the regional distribution has an OR of 2.8, as compared with the segmental or linear distribution [26]. A comparison of ORs is summarized in Table 9.

The more specific the test, the less likely an individual with a positive test will be free from disease and the greater the positive predictive value (PPV). The fine pleomorphic morphology showed lower PPV in our study, whereas heterogeneous morphology showed higher PPV in our study as compared to the study by Kim et al. Our study also showed higher PPV in regional distribution as compared to the study by Kim et al. This indicates that our study has a more specific test in terms of detecting heterogenous morphology and regional distribution; but has a less-specific test in terms of detecting fine pleomorphic morphology as compared to the study by Kim et al. [23]

Nyante et al. reported that calcifications in breast cancer were at the in situ tumor component [27]. However, our study compared the calcification patterns in benign, in situ, and invasive tumors, and we believe it does not matter whether the calcifications are in the in situ or invasive component, as our objective was to determine the calcification patterns association with different categories. Our focus was to show that in situ and invasive carcinoma have different patterns of calcification.

In our study, there were four cases of TNBC (12.1%) among 33 breast cancers. Two were in the 41–45 age group, and two were in the 61–65 and 66–70 age groups. The incidence of TNBCs in this study was similar to that in previous studies, which reported that they account for approximately 8.1–21.4% of all breast cancers [28,29,30].

In our study, 17 out of 110 patients (15.5%) with microcalcifications also had an associated breast mass. According to Chotiyano et al., mass findings are a predominant indicator of breast cancer. These malignant masses usually present as high-density lesions with irregular shapes and ill-defined or spiculated margins [31]. In our study, masses with spiculated, obscured, and microlobulation margins were seen in invasive carcinoma, and high-density masses were more commonly found in invasive carcinoma.

Our study showed that high-density masses have a higher PPV (75%) than equal-density masses (60% PPV). The masses found in mammograms with microlobulation and circumscribed margins had 100% PPVs, followed by those with spiculated margins (75% PPV) and obscured margins (63.5% PPV). These findings were similar to those reported by Liberman et al. [24].

There were several limitations to this study. First, this study was conducted during the COVID-19 pandemic period, during which Malaysia implemented the Malaysia Government Movement Control Order from 18 March 2020 to 31 December 2021. These measures included restrictions on movement, assembly, and international travel, as well as the closure of business, industry, government, and educational institutions to curb the spread of COVID-19. As a result of these restrictions, the availability of participants for our study was significantly limited. Specifically, all outpatient mammogram screening studies were temporarily withheld for several months. During this time, mammogram studies were only conducted on symptomatic women. Additionally, the screening program resumed with a limited number of appointments due to the deployment of almost half of the working forces to manage COVID-19 wards, emergency department rotations, and quarantine centers. Given these challenging circumstances, our study faced inherent limitations in terms of the available population size. Despite the constraints, we believe that our findings provide valuable insights into the specific population of symptomatic women during the COVID-19 pandemic. However, we acknowledge the importance of expanding the study population in future research to obtain more comprehensive results.

Second, it included a relatively small number of patients presenting at a single center; therefore, the results cannot be generalized. Third, in view of the retrospective design, case selection bias possibly occurred because we included only patients who underwent imaging-guided biopsies or surgeries for suspicious microcalcifications. Fourth, unavoidably uneven suspicious microcalcification distributions were included in this study, with a high incidence of fine pleomorphic morphology and grouped distribution and a low incidence of the other morphologies and distributions.

Additionally, we took the opportunity to expand upon the limitations of our study and acknowledge the need for future research to address these limitations. This includes considerations for larger sample sizes, multicenter collaborations, and further investigations into the associations between microcalcification patterns and specific breast cancer subtypes.

## 5. Conclusions

Our study showed that fine linear or branching pleomorphic, heterogeneous microcalcification morphologies, and segmental, regional, and linear distributions are associated with a higher risk of DCIS. Heterogeneous calcifications with regional distributions were associated with a higher risk of invasive carcinoma. Our study also found that microcalcifications associated with masses, particularly high-density lesions with obscured or spiculated margins, had high breast malignancy PPVs.

## Figures and Tables

**Figure 1 diagnostics-15-01687-f001:**
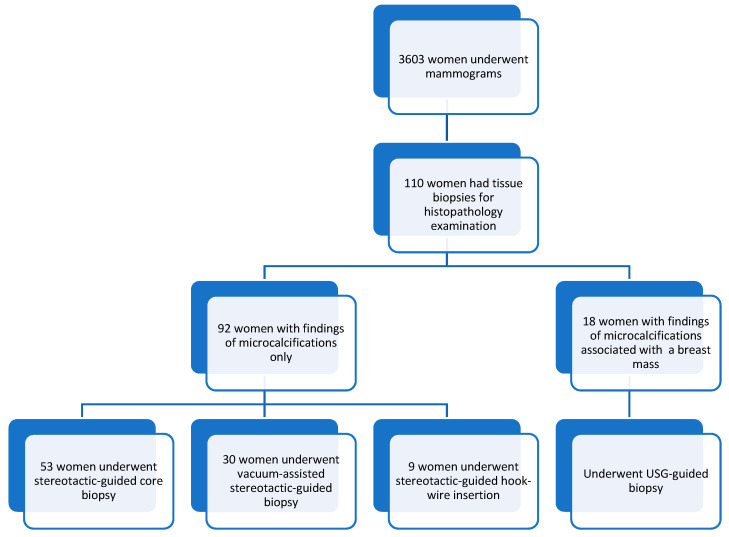
Sample distribution.

**Figure 2 diagnostics-15-01687-f002:**
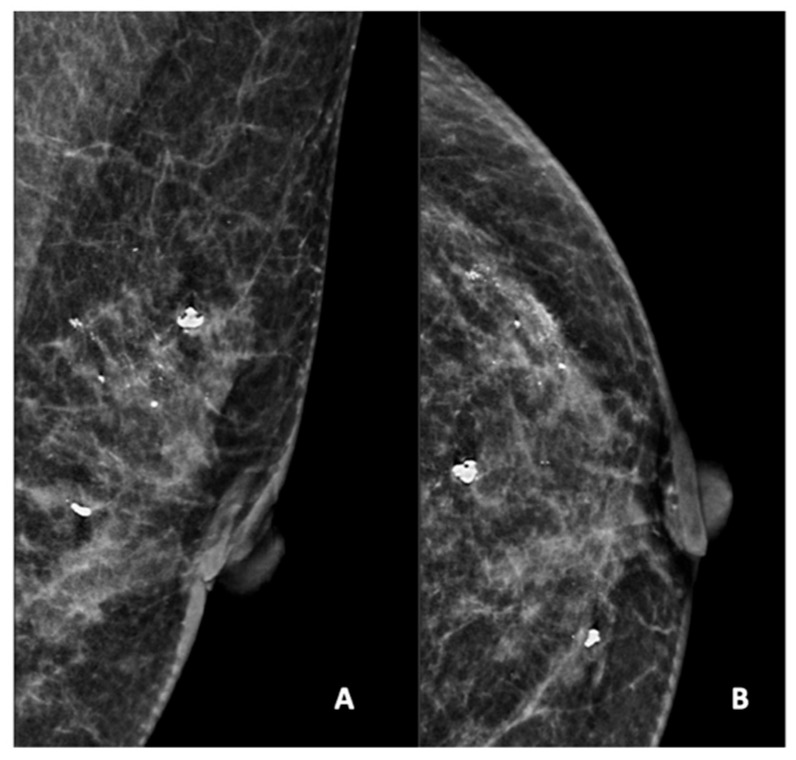
Mammogram study of a left breast in (**A**) mediolateral (MLO) view and (**B**) craniocaudal (CC) view, showing amorphous calcifications in a segmental distribution in the upper outer quadrant, reported as a BI-RADS 4 lesion. Given these calcifications, a stereotactic-guided core biopsy was performed; the HPE showed DCIS. There were also benign coarse calcifications in the upper outer and mid-inner quadrants.

**Figure 3 diagnostics-15-01687-f003:**
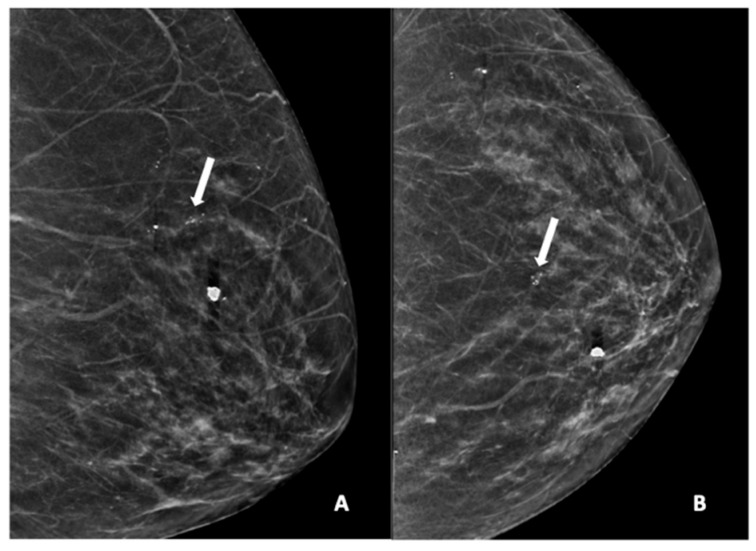
Mammogram study of a left breast in (**A**) MLO view and (**B**) CC view, showing a group of pleomorphic calcifications in the upper mid-quadrant (arrow), reported as BI-RADS 4 lesion. These calcifications were HPE-proven DCIS. There were also benign coarse calcifications in the upper inner quadrant.

**Figure 4 diagnostics-15-01687-f004:**
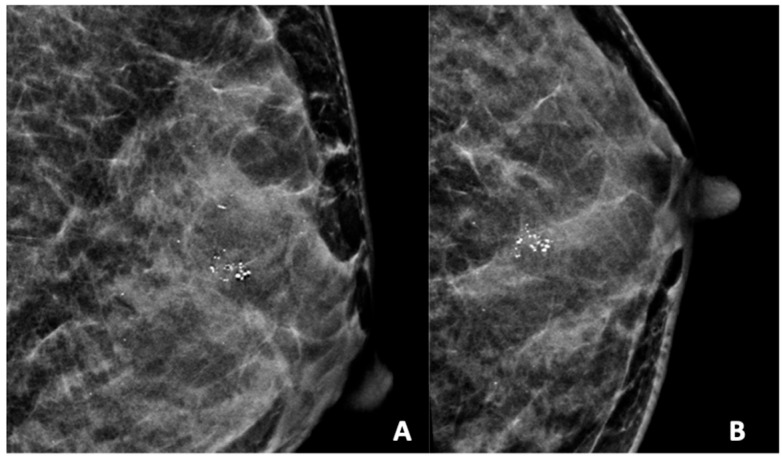
Mammogram study of a left breast in (**A**) MLO view and (**B**) CC view, showing a group of heterogeneous microcalcifications in the upper mid-quadrant, reported as BI-RADS 4B. These calcifications were HPE-proven DCIS.

**Figure 5 diagnostics-15-01687-f005:**
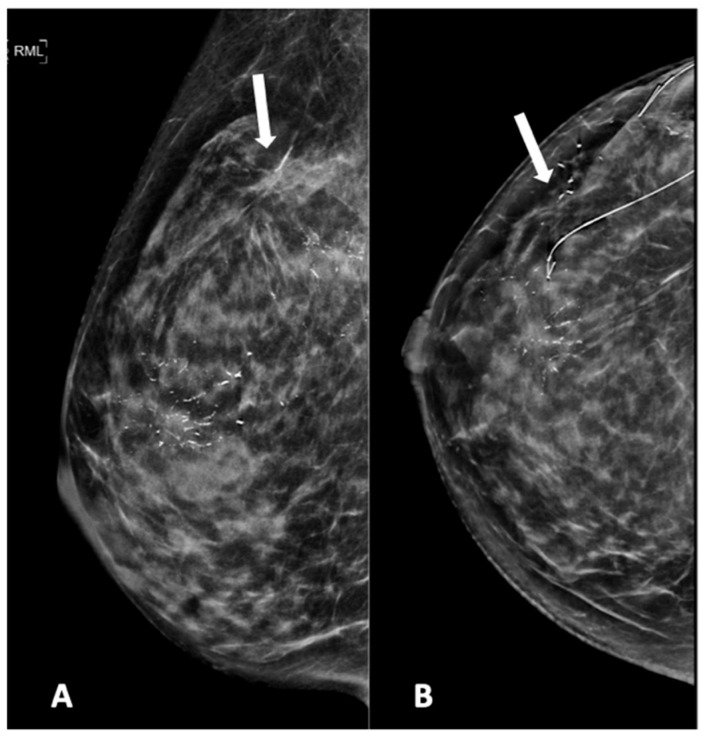
Mammogram study of a right breast in (**A**) MLO view and (**B**) CC view, showing branching pleomorphic calcifications in a segmental distribution in the upper outer-to-mid quadrant, reported as a BI-RADS 4C lesion. A stereotactic-guided hook-wire insertion was performed (hook-wire in CC view), and an HPE of the wide local excision showed invasive carcinoma. We noted an architectural distortion in the upper outer quadrant (arrow) from a previous surgery.

**Figure 6 diagnostics-15-01687-f006:**
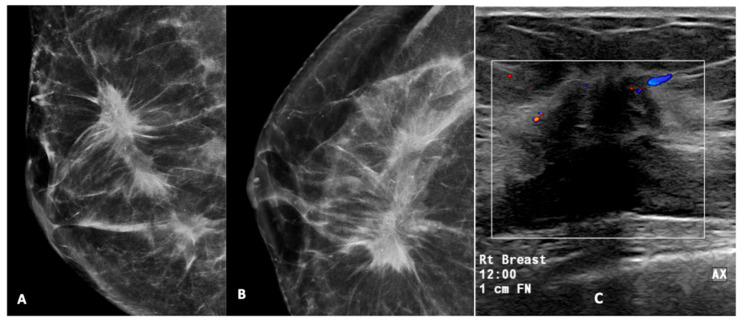
High-density mass with spiculated margin in the upper outer quadrant, associated with pleomorphic calcifications (**A**,**B**). A USG of this lesion (**C**) showed a hypoechoic mass with a spiculated margin, posterior acoustic shadowing, and internal vascularity on Doppler. This was reported as a BI-RADS 5 lesion. USG-guided biopsy was performed and was found to be HPE-proven invasive carcinoma. (Rt breast—right breast, 12.00—12 o’clock position, 1 cm FN—1 cm from nipple). The box represent the lesion and colour area represent blood flow.

**Table 2 diagnostics-15-01687-t002:** Demographic distributions of patients, presenting symptoms, and types of procedures.

Characteristics		No. of Patients	%
**Ethnicity**			
Malay		62	56.4
Chinese		31	28.2
Indian		12	10.9
Others		5	4.5
** Total**		**110**	**100**
**Symptoms**			
Palpable lesion		15	51.7
Skin changes		2	6.9
Nipple retraction		2	6.9
Nipple discharge	Bloody	4	13.8
	Serous	1	3.4
Palpable axillary nodes		5	17.2
** Total**		**29**	**100**
**Procedures**			
Stereotactic-guided core biopsy		53	48.2
Vacuum-assisted stereotactic-guided biopsy		30	27.3
Stereotactic hook-wire insertion		9	8.2
Ultrasound-guided biopsy		18	16.4
** Total**		**110**	**100**

**Table 3 diagnostics-15-01687-t003:** Histopathological examination (HPE) results based on BI-RADS categories.

BI-RADS Category	HPE Results (No. of Cases/%)		Estimated Malignancy Rate Based on ACR-BIRADS (%)	Total Cases
	Benign	Malignant		
3	2 (100%)	0 (0%)	<2	2
4	68 (68.7%)	31 (31.3%)	2–95	99
5	1 (11.1%)	8 (88.9%)	>95	9
Total	71	39	-	110

**Table 4 diagnostics-15-01687-t004:** Histopathological examination (HPE) results based on the morphology and distribution of the microcalcifications.

Types and Distributions of Microcalcifications	Fine Pleomorphic	Mixed Micro- and Macrocalcifications	Amorphous	Fine Linear or Branching Pleomorphic	Increasing Calcifications	Course Heterogeneous	Total	Group	Segmental	Regional	Linear	Total
HPE												
Ductal carcinoma in situ (DCIS)	15	2	1	3	0	2	23	15	4	3	1	23
Invasive carcinoma	13	0	1	1	0	1	16	13	1	1	1	16
Benign breast tissue	19	4	4	1	1	1	30	29	1	0	0	30
Fibrocystic changes	5	1	3	0	3	0	12	12	0	0	0	12
Fibroadenomatous change	5	1	0	0	1	0	7	6	0	1	0	7
Fibrotic breast tissue	5	1	0	0	0	0	6	6	0	0	0	6
Ductal hyperplasia	3	1	0	1	0	0	5	3	1	1	0	5
Intraductal papilloma	2	1	0	0	0	0	3	3	0	0	0	3
Sclerosing adenosis	4	0	1	0	0	0	5	4	0	1	0	5
Other *	1	0	2	0	0	0	3	3	0	0	0	3
**Total**	**72**	**11**	**12**	**6**	**5**	**4**	**110**	**94**	**7**	**7**	**2**	**110**

* Other: Benign diseases, e.g., columnar cell hyperplasia, apocrine metaplasia, and fat necrosis.

**Table 5 diagnostics-15-01687-t005:** PPV of malignancy according to the morphology and distribution of microcalcifications.

Descriptor	Benign	Malignant		Total	PPV (%)
		DCIS	Invasive Carcinoma		
**Morphology**					
Fine pleomorphic	44	15	13	72	38.9
Amorphous	10	1	1	12	16.7
Mixed micro- and macrocalcifications	9	2	0	11	18.1
Fine linear or branching pleomorphic	2	3	1	6	66.7
Increasing calcifications	5	0	0	5	0
Heterogeneous	1	2	1	4	75.0
Total	71	23	16	110	
**Distribution**					
Group	66	15	13	94	29.8
Segmental	2	4	1	7	71.4
Regional	3	3	1	7	57.1
Linear	0	1	1	2	100
Total	71	23	16	110	

**Table 6 diagnostics-15-01687-t006:** Association between the morphology and distribution of microcalcifications and breast malignancy.

Descriptor	DCIS		Invasive Carcinoma		Triple Negative	
	Odds Ratio	95% Confidence Interval (CI)	Odds Ratio	95% Confidence Interval (CI)	Odds Ratio	95% Confidence Interval (CI)
**Morphology**						
Fine pleomorphic	0.8	0.331–2.17	1.7	0.508–5.685	1.8	0.169–19.893
Amorphous	0.7	0.141–3.390	0.5	0.6–4.189	1.1	1.003–1.315
Mixed micro- and macrocalcifications	0.8	0.156–3.867	0.6	0.067–4.7	-	-
Fine linear or branching pleomorphic	3.9	0.744–21.004	1.2	0.129–10.878	2.9	0.223–37.345
Heterogeneous	3.8	0.509–28.650	2.0	0.197–20.746	1.1	1.003–1.315
Increasing calcifications	-	-	-	-	-	-
**Distribution**						
Group	0.2	0.067–0.627	0.4	0.122–1.585	0.4	0.046–3.181
Segmental	5.5	1.146–26.718	0.9	0.110–8.707	2.9	0.223–27.345
Regional	2.9	0.608–14.102	2.5	0.449–14.4	1.2	1.003–1.342
Linear	3.7	0.223–61.374	6.2	0.368–104.532	9.3	0.457–190.626

**Table 7 diagnostics-15-01687-t007:** PPV of malignancy based on mammographic mass features.

Descriptor	Benign	Malignant		Total	PPV (%)
		DCIS	Invasive Carcinoma		
**Mass density (in relation to attenuated fibroglandular tissue)**					
High density	3	1	8	12	75.0
Equal density	2	3	0	5	60.0
**Total**	**5**	**4**	**8**	**17**	
**Mass margin**					
Mass with obscured margin	4	3	4	11	63.6
Mass with spiculated margin	1	0	3	4	75.0
Mass with microlobulation margin	0	0	1	1	100
Mass with circumscribed margin	0	1	0	1	100
**Total**	**5**	**4**	**8**	**17**	

**Table 8 diagnostics-15-01687-t008:** Comparison of the PPVs of the microcalcification features in this study and other studies.

Features	PPV (%)			
	Current Study	ACR-BIRADS	Kim et al. [23]	Liberman et al. [24]
Morphology				
Fine pleomorphic	38.9	29	63.2	-
Amorphous	16.7	20.0	7.9	-
Fine linear or branching pleomorphic	66.7	70.0	100	-
Heterogeneous	75.0	15.0	17.8	-
Distribution				
Regional	57.1	26.0	8.8	46
Linear	100	60.0	87.5	68
Segmental	71.4	62.0	63.6	74
Group	29.8	Considered benign or suspicious based on the morphology of each group	14.3	36

**Table 9 diagnostics-15-01687-t009:** Comparison of the odds ratios and CIs of microcalcification features associated with invasive carcinoma in this study and Lilleborge et al.

Features	Our Study		Lilleborge et al. [26]	
	Odds Ratio	95% Confidence Interval (CI)	Odds Ratio	95% Confidence Interval (CI)
**Morphology**				
Fine linear or branching pleomorphic	1.2	0.129–10.878	20.0	2.5–158.9
Amorphous or coarse heterogeneous	0.5/2.0	0.6–4.189/0.197–20.746	1.6	0.8–3.5
**Distribution**				
Regional	2.5	0.449–14.4	2.8	1.0–8.2
Group	0.4	0.122–1.585	1.9	0.7–5.0

## Data Availability

The original contributions presented in this study are included in the article. Further inquiries can be directed to the corresponding author.

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
