# Peer review of "Association Between Microcalcification Patterns in Mammography and Breast Tumors in Comparison to Histopathological Examinations"

_diagnostics, 2025, doi:10.3390/diagnostics15131687_

Round 1
Reviewer 1 Report (Previous Reviewer 1)
Comments and Suggestions for Authors
This is a correct study which is well presented. The illustrations are good, the text is well written.
I expected a more detailed pathological correlation. For example a linear calcifications that fills a duct, a group / cluster that fills a distended lobule, etc. If not from own histopathology than in the discussion from the literature.
Some terms are not originally from the BIRADS, as crushed stone like, powdery, they are part of the Tabar system. The same is true regarding the calcifications filling a lobe that is related to the sick lobe theory. This could also be discussed,
Author Response
Reviewer 1
This is a correct study which is well presented. The illustrations are good, the text is well written.
Reply: Thanks for the comment.
I expected a more detailed pathological correlation. For example a linear calcifications that fills a duct, a group / cluster that fills a distended lobule, etc. If not from own histopathology than in the discussion from the literature.
Reply:
Thanks for the comment. We have added a few articles into the discussion. Also see the reply on the next questions.
1) Swamy R. Histological correlation of mammographically detected breast calcifications – A need for rational protocols. Diagnostic Histopathology 2009;15(12):582-588.
2) Maguire JK et al. Calcification in breast histopathology. Diagnostic Histopathology (article in press).
The study showed the most common microcalcification in benign breast lesions are within involutional lobules, columnar cell change, fat necrosis and fibrocystic change. Microcalcification could be seen in about 80% of DCIS, even without a mass. While, in malignant calcification, it is described as fine pleomorphic or fine linear/branching and usually range in size from 100 to 300 µm.
Amorphous calcifications linear or branching calcifications and custering of calcifications have been discussed. See Page 4 and 5.
Some terms are not originally from the BIRADS, as crushed stone like, powdery, they are part of the Tabar system. The same is true regarding the calcifications filling a lobe that is related to the sick lobe theory. This could also be discussed,
Reply. Thank you for pointing out. The description in Table 1 for amorphous calcifications and fine pleomorphic calcifications has been amended.
Added in Page 4, line 137
Amorphous calcification
It is also called tiny or hazy microcalcifications, about 200 to 300 µm, they are less conspicuous than the other microcalcifications and require technically optimised mammograms.
Added reference 18
Henrot, P., et al. (2014). "Breast microcalcifications: the lesions in anatomical pathology." Diagnostic and interventional imaging 95(2): 141-152.
Added in Page 5, line 137
Fine pleomorphic calcifications
It is usually more conspicuous than the amorphous forms of calcifications. Neither typically benign nor typically malignant irregular calcifications with varying sizes and shapes. It is typically more prominent as compared to the amorphous forms. Irregular calcifications of different sizes and shapes that are neither distinctly benign or malignant [35].
Added reference 19
Hofvind, S., et al. (2011). "Mammographic morphology and distribution of calcifications in ductal carcinoma in situ diagnosed in organized screening." Acta Radiologica 52(5): 481-487.
All references were amended accordingly.

Reviewer 2 Report (New Reviewer)
Comments and Suggestions for Authors
The paper of Park et al AJR 213, 710-715 is missing.
The number of cases is very small precluding meaningfull conclusions in some of the evaluated groups.
The experience of the radiologists analyzing the mammograms should be better described (number of radiologists, years of expertise).
In the results and in the discussion the limitations of the small numbers should be clearly stated (for example for the groups with less than 10 total cases in table 3,4,5,7).
The significant characters in Table 6 should be reduced. Three or more significant places are not adequate in case of the small numbers.
Abbreviations like HCTM, USG should be explained at the first mentioning (line 74, 75).
In line 77 performance should be replaced by sensitivity since ultrasound improves sensitivity but lowers specificity.
In Fig 1. stereotactic guided biopsy should be explained (was it core biopsy?).
In Table 2 a line asymptomatic should be included. Here also the stereotactic guided biopsy should be explained (probably core biopsy?)
Table 3. HPE should be explained in the table
The microcalcification types should be confined to and used as in the BIRADS 5 characterization (Table 1). Combined calcifications should be grouped according to the most suspicious micro calcification content. Increasing calcifications should be grouped into their micro calcification character and the increase be treated as an additional characterization (line 144,155, table 4).
Fig 4 shows micro- not macro calcifications. The BIRADS categorization should be included.
Fig 5 the BIRADS categorization should be included
Table 8 and 9 the numbers of the reference list should be added.
The discussion should also reflect on the small numbers that preclude far reaching conclusions (line 217,lines 233-239 etc).
The statistical excursion (lines268-280) should be dropped.
Lines 281-286 are not relevant in view of the results presented and the small numbers.
A long discussion of triple-negative breast cancers based on only 4 cases is irrelevant and should be dropped (line 287-303).
Author Response
Reviewer 2
The paper of Park et al AJR 213, 710-715 is missing.
Reply: It has been removed.
The number of cases is very small precluding meaningfull conclusions in some of the evaluated groups.
Reply: Our study recruited 3603 patients; however, only 110 women fulfilled the inclusion criteria of having microcalcifications. However, majority of them fell into BI-RADS 4, leaving only 2 patients in BI-RADS 3 and 9 patients in BI-RADS 5. Therefore, it can be concluded that most microcalcifications were detected at earlier stages before reaching BI-RADS 5.
The experience of the radiologists analyzing the mammograms should be better described (number of radiologists, years of expertise).
Reply: Thanks for the comments. The information on the experience of the radiologists have been added in ‘Study Methodology’, added in Page 4, lines 120-124.
The imaging findings were characterized by radiologists with more than 10 years of experience with special interest in breast imaging and reading mammograms according to the American College of Radiology Breast-Imaging Reporting and Data System (ACR BI-RADS) Atlas, 5th Edition.
In the results and in the discussion the limitations of the small numbers should be clearly stated (for example for the groups with less than 10 total cases in table 3,4,5,7).
Reply: Thank you for the comments. The changes have been made.
Added in the Results section: Page 7, Lines 163 – 170
Among 3603 patients, only 110 women fulfilled the inclusion criteria of having microcalcifications. Most of these women were classified as BI-RADS 4, while 2 patients fell into the BI-RADS 3 category, and 10 patients fell into BI-RADS 5 category. This finding indicates that most microcalcifications were caught in the earlier stage before advancing to BI-RADS 5.
However, the malignancy rate among BI-RADs 5 lesions is lower as compared to the reference malignancy rate (>95%) according to BI-RADS. This discrepancy may have been caused by the small sample size and early detection, which limits the representativeness of the findings.
And also added in Page 12 in Discussion section: Lines 232-235
This lower observed malignancy rate of 88.9% may be attributed to the small sample size, which is a limitation in this study. With a larger sample size, the malignancy rate may closely align with the reference malignancy rates according to the ACR BI-RADS Atlas 2013.
This also could be attributable to the early detection rate in our hospital.
The significant characters in Table 6 should be reduced. Three or more significant places are not adequate in case of the small numbers.
Reply: There are four associations in Table 6 that show significant findings (Confidence Index (CI) does not include 1). We are aware that the significant findings may not be reliable due to small sample sizes; however, this can be detected by the wide confidence intervals (for example 1.146–26.718). Therefore, the confidence interval values provided in Table 6 gives information that the findings should be interpreted cautiously. The reasons for small numbers have also been explained in the study limitations. Hope this is reasonable to you.
Abbreviations like HCTM, USG should be explained at the first mentioning (line 74, 75).
Reply: Thank you for pointing out. The abbreviations mentioned have been expanded to its full form. See Page 3.
In line 77 performance should be replaced by sensitivity since ultrasound improves sensitivity but lowers specificity.
Reply: Thank you for pointing out. The changes have been made in Page 3.
In Fig 1. stereotactic guided biopsy should be explained (was it core biopsy?).
In Table 2 a line asymptomatic should be included. Here also the stereotactic guided biopsy should be explained (probably core biopsy?)
Reply: Apologies, we have missed out the ‘core’ in stereotactic guided core biopsy. Changes have been made in Figure 1 and Table 2. See Page 7.
Table 3. HPE should be explained in the table
Reply: Thanks for the comment. The full form of HPE has been expanded in Table 3. Histopathological examination (HPE) results based on BI-RADS categories.
The microcalcification types should be confined to and used as in the BIRADS 5 characterization (Table 1). Combined calcifications should be grouped according to the most suspicious micro calcification content. Increasing calcifications should be grouped into their micro calcification character and the increase be treated as an additional characterization (line 144,155, table 4).
Reply: Mixed micro-macro calcifications and increasing calcifications were removed. The column ‘heterogeneous’ has been renamed as coarse heterogenous. See Page 7, Line 152-153.
Fig 4 shows micro- not macro calcifications. The BIRADS categorization should be included.
Reply: Thank you for the comment. Changes have been made accordingly. See Page 11.
Fig 5 the BIRADS categorization should be included
Reply: Thanks. Changes have been made accordingly. See Page 12.
Table 8 and 9 the numbers of the reference list should be added.
Reply: Thanks. Changes have been made accordingly. See Page 13.
The discussion should also reflect on the small numbers that preclude far reaching conclusions (line 217, lines 233-239 etc).
Reply: Thank you for the comments. We have added more discussion.
See Page 12. Lines 232-235
This lower observed malignancy rate of 88.9% may be attributed to the small sample size, which is a limitation in this study. With a larger sample size, the malignancy rate may closely align with the reference malignancy rates according to the ACR BI-RADS Atlas 2013
See Page 12. Lines 249-251
However, these findings cannot be generalized due to the small sample size. Further studies involving larger cohorts should be conducted to assess the validity of these findings.
The statistical excursion (lines 268-280) should be dropped.
Lines 281-286 are not relevant in view of the results presented and the small numbers.
Reply: Thanks for the comments. Changes have been made accordingly. The irrelevant sentences were removed. See Page 14. Line 290 and 309.
A long discussion of triple-negative breast cancers based on only 4 cases is irrelevant and should be dropped (line 287-303).
Reply: Changes have been made accordingly. The irrelevant sentences were removed. See Page 14. Line 318.

Round 2
Reviewer 2 Report (New Reviewer)
Comments and Suggestions for Authors
Line 32: Data of 110 of 3603 women with presence of calcifications on digital breast tomography mammogram were subjected to stereotac- tic/ultrasound (USG) guided biopsies, and hook-wire localization excision procedures and fulfilled the inclusion criteria were collected
This sentence must be reformulated:
110 out of 3603 women had microcalcification of BIRADS 3 or higher and were subjected to stereotactic/ultrasound (USG) guided biopsies, and hook-wire localization excision procedures.
Line 109: were there no benign microcalcifications (BIRADS 2) that did not need biopsy?
Fig 1: what were the exclusion criteria? Should be included in the methods section, if any or should be dropped in the figure
Line 107: in the study population it should be described that a significant number of these women were symptomatic (29 from 110 according to table 2 and text)
Line 160: not „all BIRADS 3 lesions“, better „the two BiRADS 3 lesions“
Line 166: The sentence beginning with „This finding indicates….“ Should be dropped. Microcalcifications usually do not progress through BIRADS categories!
Line 229: Not „all BIRADS 3 lesions“, Instead „the two BIRADS 3 lesions
Line 237: „One subject had a BI-RADS 5 lesion“ The authors probably meant a BIRADS 5 lesion with benign histopathology?
Regarding the small sample size throughout the paper not more than 1 decimal digits should be given also with the ORs.
For example line 270 NOT „(OR: 5.533, 95%; CI: 1.146–26.718)“ BETTER „(OR:5.5; 95% CI 1.1-26.7)“ the additional digits are meaningless and hamper easy reading. This refers to all text, figures and tables.
Author Response
Reply to reviewer’s comments:
1) Line 32: Data of 110 of 3603 women with presence of calcifications on digital breast tomography mammogram were subjected to stereotac- tic/ultrasound (USG) guided biopsies, and hook-wire localization excision procedures and fulfilled the inclusion criteria were collected
This sentence must be reformulated:
110 out of 3603 women had microcalcification of BIRADS 3 or higher and were subjected to stereotactic/ultrasound (USG) guided biopsies, and hook-wire localization excision procedures.
Reply: Thank you for reword this sentence. It has been corrected. See line 32.
2) Line 109: were there no benign microcalcifications (BIRADS 2) that did not need biopsy?
Reply: Yes, only 110 cases of a total 3603 women who underwent mammogram had microcalcifications. The rest did not need biopsy.
3) Fig 1: what were the exclusion criteria? Should be included in the methods section, if any or should be dropped in the figure
Reply: Thank you for the comment. We have changed it to the following (see below) in figure 1.
Correction: 110 women had tissue biopsies for histopathology examination
4) Line 107: in the study population it should be described that a significant number of these women were symptomatic (29 from 110 according to table 2 and text)
Reply: Thank you for pointing out this important information. We have added the sentence below as suggested in line 109.
Correction: A proportion of these women were symptomatic (29/110, 26.4%) (Table 2).
5) Line 160: not „all BIRADS 3 lesions“, better „the two BiRADS 3 lesions“
Reply: We have corrected the sentence as suggested in line 158.
Correction: The two BI-RADS 3 lesions were HPE-proven benign lesions.
6) Line 166: The sentence beginning with „This finding indicates….“ Should be dropped. Microcalcifications usually do not progress through BIRADS categories!
Reply: Thank you for the comment. We agree and have removed the sentence (line 164).
Correction: This sentence was removed - This finding indicates that most microcalcifications were caught in the earlier stage before advancing to BI-RADS 5.
7) Line 229: Not „all BIRADS 3 lesions“, Instead „the two BIRADS 3 lesions
Reply: We have corrected the sentence as suggested in line 225.
Correction: In our study, two BI-RADS 3 lesions were HPE-proven benign lesions,…
8) Line 237: „One subject had a BI-RADS 5 lesion“ The authors probably meant a BIRADS 5 lesion with benign histopathology?
Reply: Thank you for pointing out the error. We have corrected it in line 233.
Correction: One subject with benign histopathology had a BI-RADS 5 lesion.
9) Regarding the small sample size throughout the paper not more than 1 decimal digits should be given also with the ORs.
For example line 270 NOT „(OR: 5.533, 95%; CI: 1.146–26.718)“ BETTER „(OR:5.5; 95% CI 1.1-26.7)“ the additional digits are meaningless and hamper easy reading. This refers to all text, figures and tables.
Reply: Thank you for the comments. We have changed all the OR to one decimal place as suggested.
Correction:
- Line 265 (the segmental distribution of microcalcifications had the highest association (OR: 5.5, 95%; CI: 1.146–26.718) with DCIS, followed by the fine linear and branching pleomorphic morphology, with an OR of 3.9 and a 95% CI of 0.744–21.004.)
- Line 270 (followed by a regional distribution (OR: 2.5, CI 0.449–14.4).)
- Table 6, line 180
- Table 9, line 275
This manuscript is a resubmission of an earlier submission. The following is a list of the peer review reports and author responses from that submission.
Round 1
Reviewer 1 Report
Comments and Suggestions for Authors
This study compares the BIRADS categories of mammographic microcalcifications with the histopathological diagnosis and conclude which patters are associated with DCIS and which with invasive cancer. As the pdf of the manuscript shows formal mistakes and some parts of the manus are not visible in it, the evaluation cannot be completed and I have to propose rejection. In addition I would suggest further improvements in case of resubmission.
- A professional language editing is a must; already the title shows that the authors do not master the professional English language for such a publication.
- Most invasive breast carcinomas have an in situ component and the calcifications are most often located in the in situ part. Thus for making conclusions about the calcification patterns that are associated (rather suggestive of) with invasion should be based on the histological analysis of the location of the calcifications which is not presented.
- There are too many tables, some of them are unnecessary as the content is described in the text.
- The study population is very small.
- There are texbooks of hundreds of pages about this subject thus the findings are not novel, they are parts of the related textbooks. Thus the findings could be of value for a local scientific journal rather than for this one.
Comments on the Quality of English LanguageThere are plenty of errors in expressing the professional content, extern language editing of the entire text is proposed.
Author Response
Kindly refer to the file attached. Thank you.

Reviewer 2 Report
Comments and Suggestions for Authors
This manuscript, titled “Association of Microcalcifications Pattern in Mammography 2 With Breast Tumour in Histopathology Examination”, evaluates the pattern and distribution of mammographic microcalcifications in relation to histopathologic findings. The results found that specific patterns and distributions of microcalcifications were indeed associated with a high risk of breast cancer. Fine linear or branching pleomorphism and segmental distribution had a higher risk for DCIS, while heterogeneous patterns with linear distribution had a higher risk for invasive carcinoma. The manuscript is well organized, and the content of the manuscript is clearly stated. However, there are some aspects that could be further improved.
1. Although 3603 female patients were collected, only 110 had microcalcifications detected on mammography and included in the follow-up analysis. The limited sample size may cause data bias and may not support the results very strongly.
2. In Table 10, the comparison results of Fine pleomorphic, heterogeneous, and regional features in the current study and Kim et al. showed a large difference. In Table 11, the odds ratio of Fine linear or branching pleomorphic in current study and Lilleborge et al.’s study, also showed a great difference. Could the authors give more explanation for these results? It is suggested to describe the similarities and differences between the study of Kim et al., Lilleborge et al. and current manuscript.
3. This manuscript links descriptive findings of microcalcification on mammogram to the corresponding histopathological assessment categories, and may provide guiding suggestions for early prediction of breast cancer risk. The manuscript was mainly done by counting the values of positive predictive value (PPV) and the odd ratio of Pearson's chi-squared test. Can other evaluation metrics be included, such as significance p-value.
4. It is suggested to standardize the format of all tables.
5. the statements in the line 174 should is “There was only 1 among the 9 lesions of BI-RADS 5 was HPE-proven benign lesion”
6. for the caption of fig6, line 237, maybe some statements are missing!!
Comments on the Quality of English LanguageThe manuscript needs to be checked carefully before submitting.
Author Response

(The authors gave the same response as above.)

Round 2
Reviewer 2 Report
Comments and Suggestions for Authors
I think the manuscripts in current state can be accepted and considered for publication
Comments on the Quality of English LanguageThe language of the manuscript is clear